# A Tree-Ring-Based Precipitation Reconstruction since 1760 CE from Northeastern Tibetan Plateau, China

**Youping Chen[1], Feng Chen[1,2,3,*] and Heli Zhang[1,2,3]**

[1] Yunnan Key Laboratory of International Rivers and Transboundary Eco-Security, Institute of International Rivers and Eco-Security, Yunnan University, Kunming 650500, China; 20190012@yun.edu.cn (Y.C.); zhangheli@idm.cn (H.Z.)

[2] Key Laboratory of Tree-Ring Physical and Chemical Research of China Meteorological Administration, Institute of Desert Meteorology, China Meteorological Administration, Urumqi 830002, China

[3] Key Laboratory of Tree-Ring Ecology of Xinjiang Uigur Autonomous Region, Institute of Desert Meteorology, China Meteorological Administration, Urumqi 830002, China

\* Correspondence: feng653@yun.edu.cn

**Abstract:** Hydroclimatic conditions and related water resources change in the Tibetan Plateau is one of the main concerns for future sustainable development in China. This study presents a 254-year precipitation reconstruction from August of the previous year to June of the current year for the northeastern Tibetan Plateau based on tree-ring width data of tree-ring cores of *Picea crassifolia* from three sampling sites. The precipitation reconstruction explained 51.4% of the variance in instrumental precipitation during the calibration period 1958–2013. Dry periods with precipitation below the 254-year average value occurred during 1848–1865, 1873–1887, 1898–1923, and 1989–2003, and wet periods (precipitation above the mean) occurred during 1769–1785, 1798–1833, 1924–1938, 1957–1968, and 2004–2013. Spatial correlation analyses with the precipitation gridded dataset showed that our reconstruction contains some strong regional-scale precipitation signals for the upper Yellow River Basin. Our precipitation reconstruction also agreed in general with other dendroclimatic precipitation reconstructions from surrounding regions. In addition, reconstructed precipitation changes were consistent with the streamflow variation of the Yellow River.

**Keywords:** tree rings; precipitation reconstruction; Tibetan Plateau; Yellow River

## 1. Introduction

Precipitation and its related hydroclimatic conditions are important to the regional sustainable development because it influences agricultural production, industrial development, and freshwater supply, especially in the arid and semi-arid areas of northern China [1,2]. Climate simulation verification and flood prediction require reliable hydroclimatic information over the different time-scales [3–5]. However, short-term observation hydroclimatic records do not apply in respect of investigating past climate fluctuations. Tree rings are useful as paleoclimate proxies that extend our knowledge about past hydroclimatic information with annual resolutions and can be used to help us to capture long-term hydroclimatic fluctuations [6–10].

As the world's highest plateau with a mean altitude elevation exceeding 4500 m, Tibetan Plateau influences the Asian monsoon circulation and water transmission and distribution and becomes one of the most climate-sensitive regions in Asia [11]. Meanwhile, the harsh environment in the Tibetan plateau makes tree growth very sensitive to climate change [12–15]. Many dendrochronologists have recently developed the climate record based on the tree ring of *Qilian juniper* (*Juniperus przewalskii Kom.*) over the past millennium or even longer for the northeastern Tibetan Plateau [15–18]. However, there have been few precipitation reconstruction records based on the tree-ring of *Qinghai spruce* (*Picea crassifolia Kom.*). Meanwhile, the spatio–temporal variation of hydroclimate in the northeastern

Tibetan Plateau is very complex. Thus, more dendroclimatic reconstructions are still needed for an intensive understanding of hydroclimate change in the northeastern Tibetan Plateau [19,20].

As an important part of the Asian water tower, the northeastern Tibetan Plateau is the upstream area of the China great rivers, such as the Yellow River and Yangtze River [21]. As the main water resource for north-central China, the freshwater resources of the Yellow River account for 2% of the total freshwater resources of China and provide over 15% of the irrigation water for China's cropland. About 40% of the river flows of the Yellow River come from the headwater areas in the northeastern Tibetan Plateau [22,23]. In recent years, some researchers have made many achievements in reconstructing the streamflow series of the Yellow River based on tree rings [22–26]. However, our knowledge of the impact of climate change on streamflow of the northeastern Tibetan Plateau is still limited due to the lack of long-term hydroclimatic series [27]. Instrumental data analyses and model simulations display that the temperature has risen dramatically, and the overall precipitation has also slightly increased in the Tibetan Plateau during the last 50 years [27]. Compared with the temperature changes, the influence of hydroclimate change on the streamflow complexity has drawn more attention due to its vital role in the hydrological cycle [28]. In this paper, we present a precipitation reconstruction based on tree rings of *Picea crassifolia* for the northeastern Tibetan Plateau, China. The aims of the study were to reveal the linkages between tree-ring growth and climate parameters, to reconstruct regional precipitation series since 1760 CE in the northeastern Tibetan Plateau, and to analyze regional rainfall patterns and their associated streamflow change of the Yellow River.

## 2. Material and Methods

### 2.1. Geographical Settings and Tree-Ring Data

The study region is situated in Maixiu Town, Zeku County, Huangnan Prefecture, Qinghai Province in the northeastern Tibetan Plateau. It is the upstream area of the Yellow River and an important freshwater supply source for the communities and settlements in the nearby regions (Figure 1) [13]. The climate of the sampling region has both a cold temperature dominated by mid-latitude westerly and a plateau mountain climate, with characteristics of dry valleys, low precipitation, cold temperatures, and plenty of sunshine [13]. The most dominant tree species in the study region are spruce and juniper. The sampled tree species, *Picea crassifolia Kom*, is a common dominant species with coverage of the tree layer ranging from 50% to 70% [13].

We selected mature spruce forests with no signs of extensive logging activities for the sampling activities on the hillside facing the north in three sites (MAX, MXS, and MXX). Two cores were sampled from each tree from different directions used the increment borers at breast height. In combination, 127 cores from 54 trees were collected. Site information, including latitude and longitude, elevation, and core/tree number, is listed for three sites in Table 1. The three sampling sites are relatively close to each other (Figure 1).

**Table 1.** Statistical characteristics of the regional chronology.

| Site | Lat. (N) | Lon. (E) | Elevation (m a.s.l) | Core/Tree | Period | MSL | MS | S/N | EPS > 0.85 |
|------|----------|----------|---------------------|-----------|--------|-----|-----|-----|------------|
| RC | | | | 127/54 | 1727–2013 | 124 | 0.14 | 24.08 | 1760 |
| MAX | 35.25° | 101.84° | 3414 | 46/25 | 1727–2006 | | | | |
| MXS | 35.25° | 101.86° | 3203 | 51/16 | 1800–2013 | | | | |
| MXX | 35.33° | 101.93° | 2799 | 30/13 | 1734–2013 | | | | |

MSL: mean segment length; MS: mean sensitivity; S/N: signal-to-noise ratio; EPS: expressed population signal.

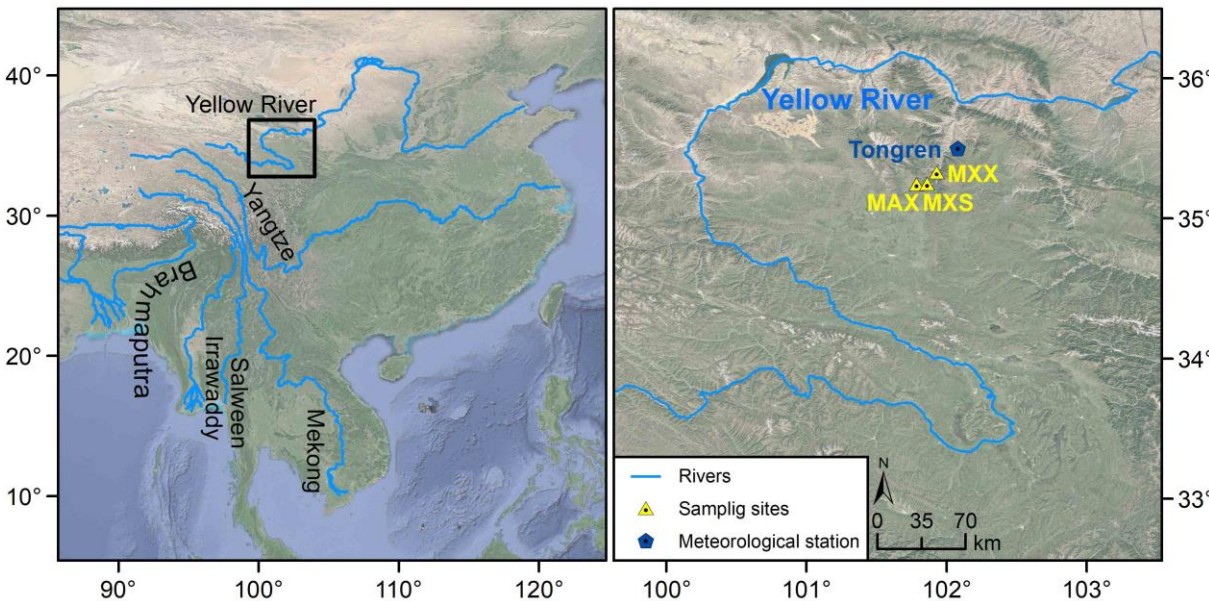

**Figure 1.** Maps of the sampling sites and the nearest Tongren climate station in the northeastern Tibetan Plateau.

The cores were air-dried, mounted on wooden holders, and polished with progressively the sandpaper (400 grit). Tree-ring widths were measured with the LINTAB 6 measuring system with a precision of 0.001 mm using the software TSAP-Win. The quality of the cross-dated tree-ring series was checked by COFECHA software [29]. Correlation coefficient between MAX and MXS of raw ring-width data is $r = 0.53$ ($p < 0.01$, 1800–2006). Correlation coefficient between MAX and MXX of raw ring-width data is $r = 0.38$ ($p < 0.01$, 1833–2006). And correlation coefficient between MXS and MXX of raw ring-width data is $r = 0.61$ ($p < 0.01$, 1833–2013). Since a high correlation was found between the raw ring-width data, we combined all raw ring-width data from three sites. To eliminate biological growth trends and possible influences of non-climatic factors from local stand dynamics and disturbance, each individual tree-ring width series were detrended with the negative exponential curve and merged into regional chronology (RC) with a bi-weight robust mean using the software ARSTAN [30]. The bi-weight robust mean reduces the influence of extreme values in the tree-ring index [30]. The ARSTAN program presents three chronologies: the standard (STD) chronology, the residual (RES) chronology and the arstan (ARS) chronology [30]. The RES chronology contains residual indices after pre-whitening, and the STD chronology through autoregressive pre-whitening, which removes autocorrelation from the original tree-ring series [30]. In our study region, the tree stands are commonly open. Therefore, we used the residual chronology (RES), which emphasizes high-frequency variations for further analysis. The mean inter-series correlation coefficient (Rbar) and the expressed population signal (EPS) and were computed for 50-year moving windows with 25-year overlaps based on per core to assess the strength and reliability of the RES chronology over time [31]. The EPS threshold of 0.85 was exceeded after 1760, when four trees were included in the dataset. A relatively high mean sensitivity (0.14) and signal-to-noise ratio (24.08) of the RC chronology show that a common environmental influence on the tree growth of *Picea crassifolia Kom.* [31] (Figure 2 and Table 1).

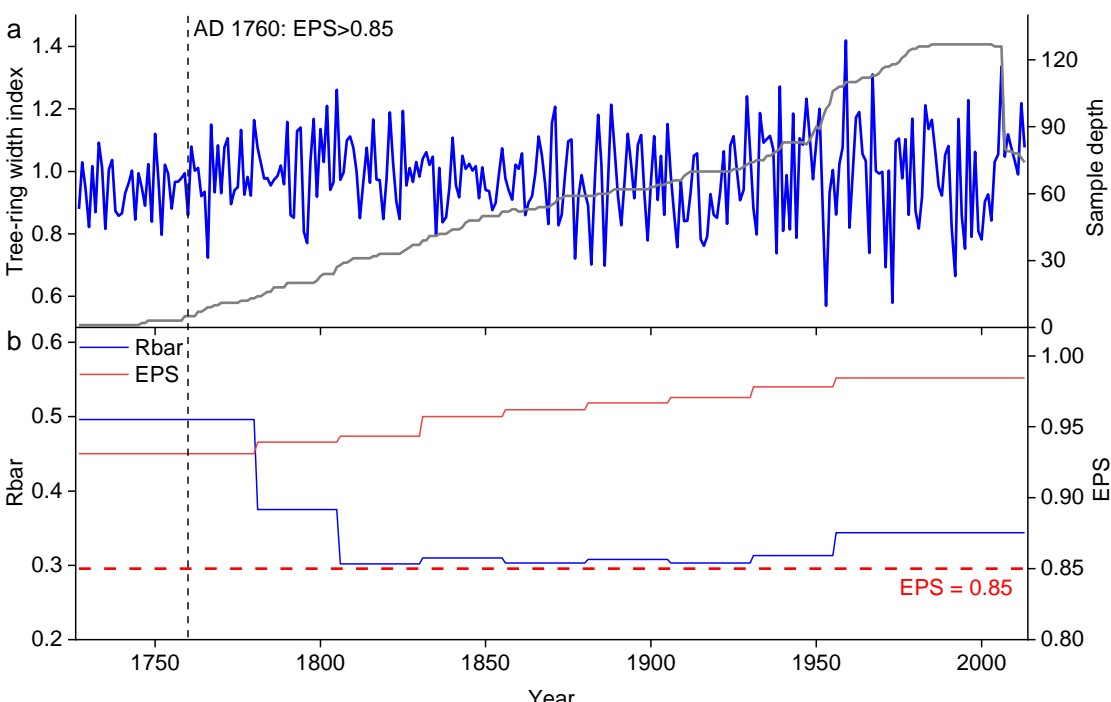

**Figure 2.** Tree-ring width residual chronology and sample depth from the study site (**a**). Running expressed population signal (EPS) and correlation coefficient (Rbar) statistics. The dashed horizontal line denotes the 0.85 cutoff value (**b**). The reliable chronology begins from the year 1760 (EPS > 0.85).

### 2.2. Climate Data and Dendroclimatic Analyses

The nearby Tongren climate station (35.31° N, 102.01° E, 2491.4 m a.s.l.) is the closest station to the sampling sites recorded an average annual temperature of 5.7 °C (1958–2013), with a mean temperature of −7.20 °C in January (the coldest month) and 16.5 °C in July (the warmest month). The average annual precipitation is 414.2 mm, and approximately 83% occurred from May to September (Figure 3). We employed average monthly temperature and total monthly precipitation (from July of the previous year to September of the current year) of the meteorological stations over the common period 1958–2013 to calibrate the ring-width climate relationships because tree growth can be affected by environmental conditions of the previous and current growing seasons [32]. Moreover, spatial correlation analyses between the residual chronology and 0.5° × 0.5° gridded precipitation (CRU TS 4.01) dataset [33] were also computed using KNMI-Climate Explorer (http://climexp.knmi.nl/ (accessed on 8 November 2020)) to identify regional climate signals in the tree-ring chronology.

The simple Pearson correlation analysis in the SPSS program was applied to indicate the climate-tree growth linkages. All statistical procedures were evaluated at a $p < 0.05$ level of significance unless otherwise noted. After identifying the highest correlation between our tree-ring chronology and the climate factors from July of the previous year to September of the current year, a linear regression model was used to develop total previous August–June precipitation reconstruction for the study region. Because the observed climate data were long enough, the split calibration and verification periods were applied to assess the goodness-of-fit of the reconstruction equation model [32]. We applied a reduction of error (RE), the coefficient of efficiency (CE), and sign test (ST) to verify the reliability and stability of our precipitation reconstruction. Positive RE and CE indicate the validity of the regression model, and the ST counts the number of disagreements and agreements between the actual and the estimated data [32]. We also compared our precipitation reconstruction with the previous tree-ring-based climate reconstructions in the northeastern Tibetan Plateau, which are controlled by similar climate conditions and

cover longer periods. We finally compared our precipitation reconstruction with previous streamflow reconstructions of the Yellow River [22] to evaluate the impact of regional precipitation on streamflow.

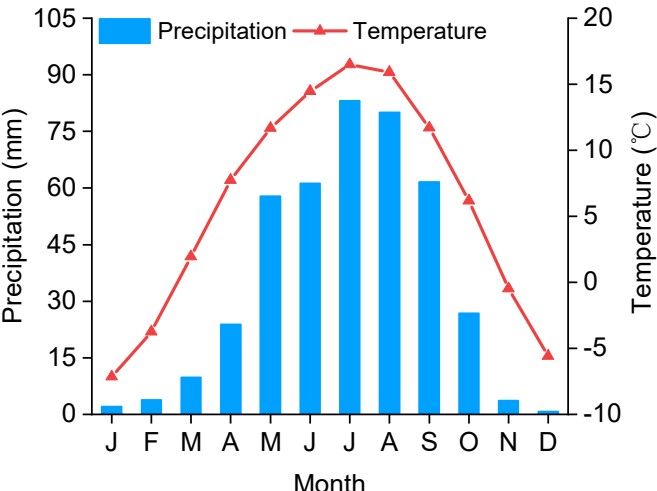

**Figure 3.** Monthly mean temperature and precipitation over the period 1958–2013 derived from Tongren meteorological station.

## 3. Results and Discussion

### 3.1. Response of Radial Growth to Climate Changes

As shown in Figure 4, the climate response results show that precipitation has stronger influences on the tree-ring width of *Picea crassifolia*. Significant positive correlations ($p < 0.01$) between the RES chronology and precipitation were found in prior September and October. After combining the months, total previous August–June precipitation showed the highest significant positive correlation with RES chronology ($r = 0.72$, $p < 0.01$). This result revealed that total previous August–June precipitation is the most important climate factor affecting tree-ring width of *Picea crassifolia* in the northeastern Tibetan Plateau.

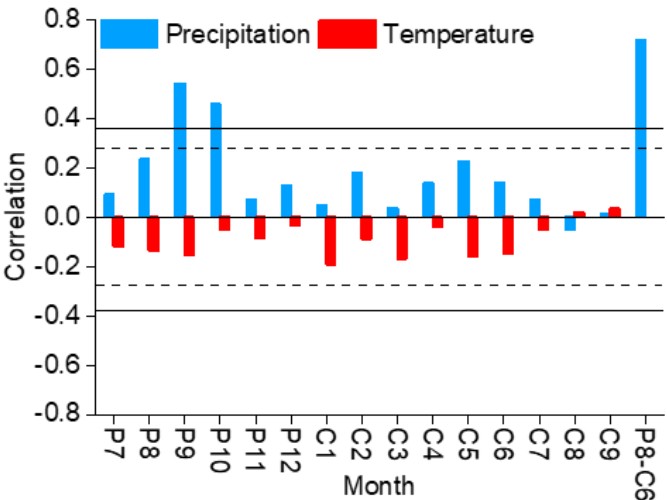

**Figure 4.** Correlation analyses between the residual chronology and monthly total precipitation and average temperature during 1958–2013. "P" and "C" represent the previous year and the current year, respectively, and the solid and dashed lines represent the 99% and 95% confidence level, respectively.

The correlation analysis indicated a robust linkage between the tree-ring width and precipitation, which confirm that moisture availability was a dominant climatic factor, which consistently explains the interannual changes in radial growth of *Picea crassifolia* in

the northeastern Tibetan Plateau. Similar results about the climate-tree growth relationship were found in the northeastern Tibetan Plateau [34–36] and other regions, especially in semi-arid Northwest China [37,38]. Tree growth is affected by the climate during the growing season and pre-growing season. During the pre-growing season, snow accumulated can insulate the soil and contribute to keeping warm soil temperatures in winter and rapid water absorption by the roots [32]. While during the growing season, sufficient precipitation supplies the important fresh water for tree growth, replenishes the lack of soil water caused by evapotranspiration, and further increases the total photosynthesis efficiency and stimulates the tree growth and division of tree cells to compose relatively wider tree rings [32].

Most of the correlations between tree growth and temperature were low negative. The low negative correlations between tree growth and mean temperature from prior July to current July reveal that temperature has no significant effect on the tree growth at these sites. These climate–growth linkages were also reported for the other areas of the central Qilian Mountains, northeastern Tibetan Plateau [19].

*3.2. Precipitation Reconstruction*

Based on the above response analysis results, the total previous August–June precipitation was reconstructed from 1760 to 2013 CE. The transfer function was designed as follows:

$$P_{86} = 232.045RCres + 98.617, \tag{1}$$

where $P_{86}$ is the reconstruction of total previous August–June precipitation and RCres is the tree-ring width chronology. The linear regression equation that was used to develop the total previous August–June precipitation reconstruction for the northeastern Tibetan Plateau passed all statistical verification tests (Table 2). The equation accounts for 51.4% ($p < 0.01$) of the actual previous August–June precipitation variance over the calibration period 1958–2013 (Figure 5a,b). The split-sample calibration-verification tests showed that the regression equation is stable through time and the reconstruction reliability is confirmed by the positive value of reduction of error (RE) and the coefficient of efficiency (CE) values. The sign-test (ST) statistic is also significant, indicating that both the observed and reconstructed values for high-frequency variations are relatively consistent.

**Table 2.** Verification statistics for the reconstruction model.

|  | Calibration (1959–1985) | Verification (1986–2013) | Calibration (1986–2013) | Verification (1959–1985) |
|---|---|---|---|---|
| $r$ | 0.762 ** | 0.675 ** | 0.675 ** | 0.762 ** |
| $r^2$ | 0.581 | 0.456 | 0.456 | 0.581 |
| RE |  | 0.450 |  | 0.571 |
| CE |  | 0.447 |  | 0.569 |
| ST |  | 24+/4- ** |  | 20+/7- * |

$r$: Person's correlation coefficient; $r^2$: explained variance; RE: reduction of error; CE: coefficient of efficiency; ST: sign-test; **: 99% confidence level; *: 95% confidence level.

*3.3. Characteristics of the Precipitation Reconstruction*

The long-term average value of total previous August–June precipitation is 327.8 mm over the last 254 years (1760–2013 CE). For the period 1760–2013, a total of 36 wet (+) and 42 dry (-) events with standard values exceeding standard deviations 1 (wet/dry) and 2 (very wet/very dry) have been indicated. Of these years, 1882, 1886, 1953, 1971, 1973, and 1992 were exceptionally dry. Drought durations were usually one year long (there were such 31 dry years), and however, in 1795–1796, 1910–1911, 1915–1917, 1991–1992, and 1999–2000, drought periods persisted over 2–3 year. The years 1805, 1939, 1959, 1967, and 2006 were very wet years. The very wet periods also were mostly one-year-long (27 wet years), and in 1793–1794, 1870–1871, 1962–1963, and 1983–1985, the wet periods were longer. The years 1959 and 1953 were the wettest and the driest, respectively.

More humid periods lasting more than 10 years occurred during 1769–1785, 1798–1833, 1924–1938, 1957–1968, and 2004–2013. Dry periods prevailed during 1848–1865, 1873–1887, 1898–1923, and 1989–2003. The number of wet periods exceeds that of dry periods during the last 254 years (Figure 5c).

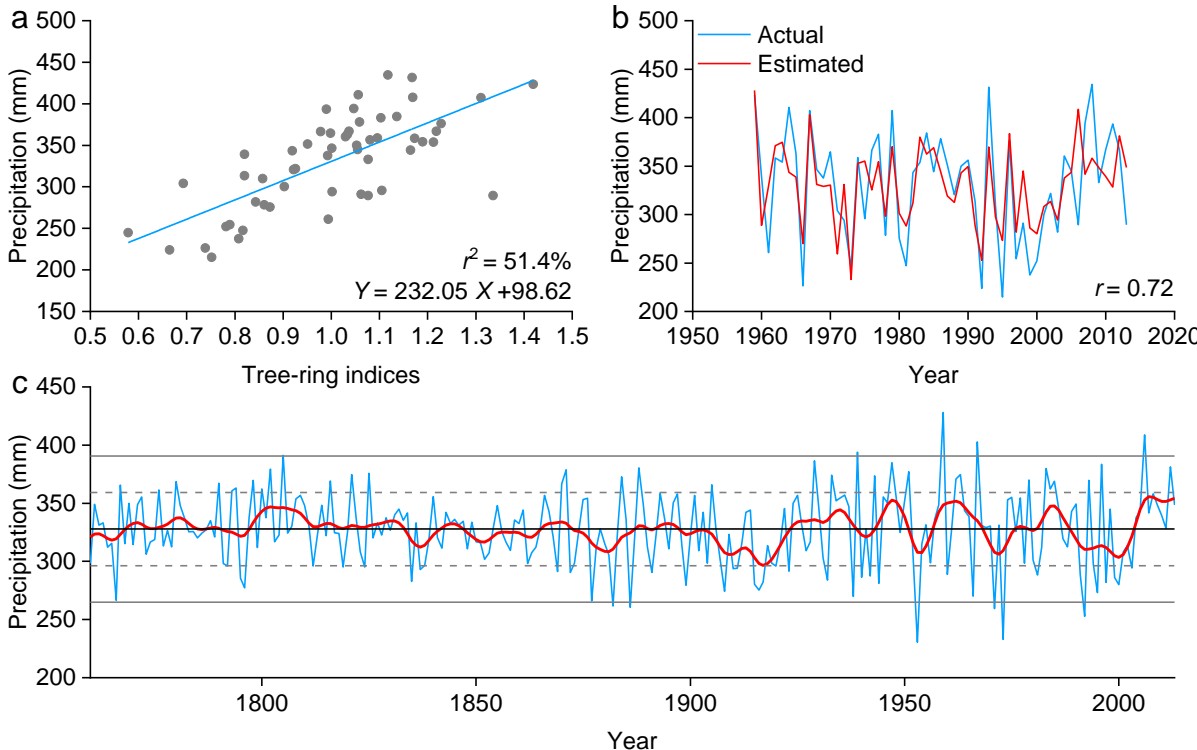

**Figure 5.** (**a**) Scatter plot of August–June total precipitation and tree-ring residual chronology with a linear fitting curve (1958–2013). (**b**) Comparison between actual and reconstructed precipitation during the common period 1958–2013. (**c**) Reconstructed total previous August-June precipitation (blue line) and 31-year smoothing (red line). The inner horizontal (dotted) lines indicate the border of one standard deviation, and outer horizontal lines indicate two standard deviations.

Dry periods prevailed during 1848–1865 and 1989–2003, and are present in tree ring reconstructions at Mt. Xinglong, which belongs to the eastern part of the Qilian Mountains [18,39]. The dry period that prevailed during 1898–1923 is also present in other dendroclimatic reconstructions from the eastern Qilian Mountains [17]. According to Zhao et al. [40], the severe drought in 1999 led to a grassland degeneration and a serious reduction in food production in northern China. Meanwhile, a large-scale drought occurred in the North China Plain during this period [41]. The wet period that prevailed during 1769—1785 and 1924—1968 are present in dendroclimatic reconstructions in the south slope of the middle Qilian Mountains and northeastern Tibetan Plateau [18]. These indicate that the moisture changes in the northeastern Tibetan Plateau have been synchronized in a large spatial and temporal range, especially during periods of drought in recent years. However, there were some unconformities in these series, which may be due to differences in microhabitats at different sampling sites or different detrending methods.

### 3.4. Linkage with Streamflow of the Yellow River

To reveal the spatial representativeness of the precipitation reconstruction further, we calculated the correlation coefficients between the reconstructed precipitation series for the period 1958–2013 using the gridded CRU TS 4.04 dataset and plotted the results using KNMI Climate Explorer (https://climexp.knmi.nl) (accessed on 8 November 2020). The reconstructed previous August–June precipitation correlated significantly with the gridded precipitation over a region covering the upper Yellow River (33–37° N, 100–115° E)

($r > 0.4$, $p < 0.01$) (Figure 6). These results indicated that our precipitation reconstruction can capture the occurrences of wet and dry events in a large area in the northeastern Tibetan Plateau, especially in the upper reaches of the Yellow River.

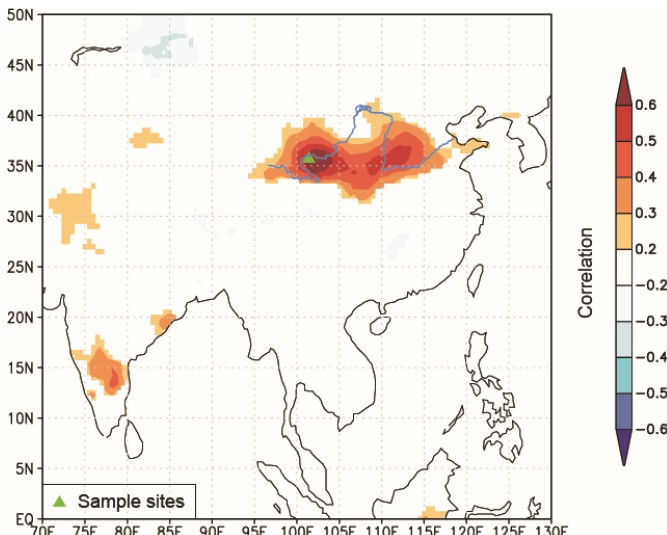

**Figure 6.** Spatial correlations between the residual chronology and the gridded previous August–June precipitation (CRU TS 4.04) for the period 1958–2013.

Precipitation is the main water resource of the Yellow River. Therefore, understanding past precipitation changes is important for estimating the state of water resources in a larger area of the Yellow River. To assess the impact of reconstructed precipitation series on flow in the reaches of the Yellow River. We used the most representative streamflow reconstruction in the middle reaches of the Yellow River [22] to compare with our precipitation reconstruction sequence (Figure 7). Correlation between the two reconstructions sequence, calculated over the common period 1760–2011 is $r = 0.34$ ($p < 0.01$). After a 21-year average smoothing, the two reconstruction sequences showed droughts in 1770–1784, 1799–1811, 1938–1951, and 1978–1988, and floods in 1833–1848, 1872–1886, and 1913–1924 accounting for 38% of the total period. The droughts in 1833–1848 are also present in the dendroclimatic reconstructions from the Yellow River headwaters and middle region [24,26], and the 1799–1811 floods are also present in tree rings from the Yellow middle region [26]. These drought and flood events are indicated by reconstructed hydroclimatic series, which supports the impact of precipitation on hydrological changes in the Yellow River.

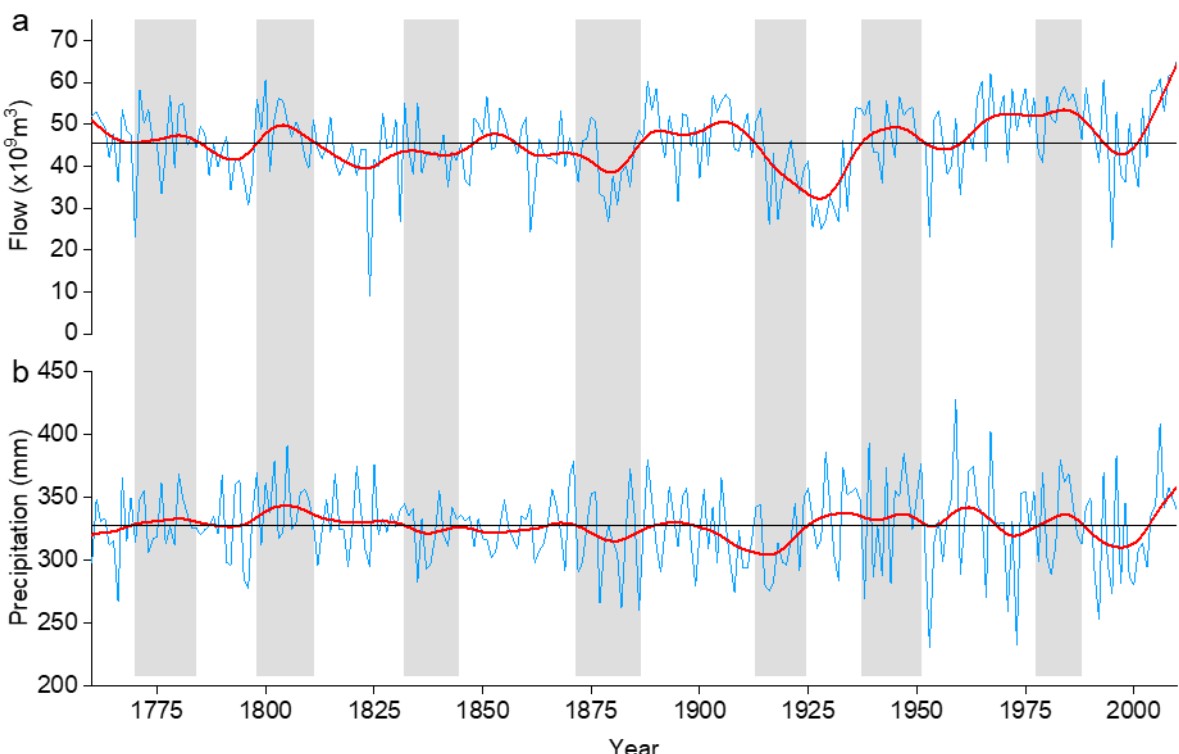

**Figure 7.** Comparison between the October–September flow reconstruction of middle Yellow River (**a**) and August–June total precipitation (**b**) over the 1760–2010 period. All series were smoothed with a 21-year low-pass filter to emphasize decadal fluctuations. The shadow means the common wet/dry periods.

## 4. Conclusions

Three local site chronologies from *Picea crassifolia Kom* in the northeast of the Tibetan Plateau were combined to form a master chronology, using a total of 127 sample cores from 54 trees. Regional tree growth proved highly sensitive to previous August–June precipitation. Previous August–June precipitation changes were reconstructed for the last 254 years (1760–2013 CE), and the precipitation reconstruction explains 51.4% of the variance in observed data over the 1958–2013 period. Our 254-year precipitation reconstruction showed good spatial representation and revealed four dry periods that occurred during 1848–1865, 1873–1887, 1898–1923, and 1989–2003. It also revealed five wet periods occurring during 1769–1785, 1798–1833, 1924–1938, 1957–1968, and 2004–2013. Meanwhile, precipitation reconstruction is consistent with the Yellow River flow changes in 1770–1784, 1799–1811, 1833–1848, 1872–1886, 1913–1924, 1938–1951, and 1978–1988, shows precipitation in this area has an important impact on the flow of the Yellow River.

**Author Contributions:** Conceived and designed the experiments: F.C. and Y.C. Performed the experiments: F.C. and Y.C. Analyzed the data: F.C. and Y.C. Contributed reagents/materials/analysis tools: F.C., Y.C., and H.Z. Contributed to the writing of the manuscript: F.C., Y.C., and H.Z. All authors have read and agreed to the published version of the manuscript.

**Funding:** This work was supported by NSFC Project (U1803341 and 32061123008) and National high-level talents special support plan.

**Data Availability Statement:** Data set available on request to corresponding authors.

**Acknowledgments:** We thank the Jinbao Li team from the Department of Geography, University of Hong Kong, Hong Kong, for providing streamflow data.

**Conflicts of Interest:** The authors declare no conflict of interest.

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
