# Peer review of "A Tree-Ring-Based Precipitation Reconstruction since 1760 CE from Northeastern Tibetan Plateau, China"

_atmosphere, doi:10.3390/atmos12040416_

Round 1

Reviewer 1 Report

This is the review of the manuscript from Youping Chen et al. considered for publication in Atmosphere [Manuscript ID: atmosphere-1105542]. Authors developed over 250 years of precipitation reconstruction for the northeastern Tibetan Plateau based on tree-ring data of Picea crassifolia. Based on the reconstruction of precipitation, the authors indicated dry and wet periods that occurred in the years 1760-2013.

 Overall, I found that the work is interesting, the presented research material is  valuable. In regions where the lack of long-term climate data, proxy series for climate reconstruction that evaluate the history of past drought play a very important role. Tibetan Plateau is such a region. The incremental series can be a very good proxy for such reconstruction and this study is an important, valuable contribution to knowledge for assessing climate change. I think, the manuscript can be interesting contribution for the readers of the journal. However, the manuscript needs improvement, especially the Methods section. I have few comments and suggestions to you.

Below I list specific comments:

  1. Regarding the title of the article: A Tree-Ring-Based Annual Precipitation Reconstruction since 1760 CE from northeastern Tibetan Plateau, China

The title of the manuscript  suggests that the research concern the reconstruction of annual precipitation,  while this study presents the reconstruction of the period from prior August to current June. Could you explain this? Maybe it would be worth changing the title, e.g. throw out the word "annual".

 2.Methods: The approaches used are appropriate. However, the description of the methodology is incomplete and most steps of the data analysis were not introduced in the Methods section at all. They are revealed to the reader just at Results section. It seems to me that the description of the methods used in the study should be more understandable to readers with less knowledge in the field of dendrochronology and climate reconstruction based on tree-rings.

Section 2.1.

  • There is no indication in this section about the definition of the reliable period. Is it only a period based on EPS threshold? If you have used this criterion, it should be indicated in this section or explained in the caption to figure 2. So, I have a question - What was the sample depth for 1760? In the Figure 2 it is approximately 6 samples/3 trees? Don't you think it's a bit too little?
  • Line 100 and/or Table 1 –RC chronology – RC abbreviation is not defined, it be should be explained

Section 2.2. Climate data and dendroclimatic analyses 

  • Lines 107-114 - You give a brief description of the climatic conditions here, but there is no information about climatic data/ variables used in the analyses.

So, I have a questions:  what climatic data were used in the study?  What was the time range of these data?  – period 1958-2013? It is not clear.  

You show it in Figure 4 (section Results) , but in my opinion these information  should be in this section. 

  • Line 118- “The correlation analysis was applied to indicate the climate-tree growth linkages” – in my opinion this information is insufficient.

For example: no information about time  interval  for  calculated the correlations coefficients,  no information how was evaluated the statistical significance of the correlations,  bootstrap methods ? This information is necessary in this section.

The next sentence is very unclear (lines 118- 122).

  • Lines 130-132 “We finally compared our precipitation reconstruction with previous streamflow reconstructions of the Yellow River to evaluate the impact of regional precipitation on streamflow” – no information about source this data

Moreover, information about the software used for calculations would also be useful.

  1. Results
  • Lines 138-139 “Total August-June precipitation showed the highest significant positive correlation with RES chronology” - I have a question- Have you analysed the correlations for other periods? , for example, correlations between annual rainfall, or maybe other periods, for example, August-October of the previous year. If you have not analysed other periods, please write why you chose the August-June period.
  • Figure 4 – please explain abbreviation P i C on the OX axis.
  • Lines 162-166 “Low winter temperatures can lead to thicker snow cover, further delay initiation of growth in the following spring which reduced earlywood width [30]”. –Could you explain this sentence in relation to your results. After all, in your research correlations between winter temperature and temperature are negative.

3.4. Linkage with steamflow of the Yellow River- typo error – should be streamflow

Best regards,

Author Response

Point 1: The title of the manuscript suggests that the research concern the reconstruction of annual precipitation, while this study presents the reconstruction of the period from prior August to current June. Could you explain this? Maybe it would be worth changing the title, e.g. throw out the word "annual".

Response 1: Yes, this is mistake, and have thrown out the word "annual".

Point 2: Section 2.1. There is no indication in this section about the definition of the reliable period. Is it only a period based on EPS threshold? If you have used this criterion, it should be indicated in this section or explained in the caption to figure 2. So, I have a question - What was the sample depth for 1760? In the Figure 2 it is approximately 6 samples/3 trees? Don't you think it's a bit too little?

Response 2: Yes, I have added the relevant sentence. We define the reliability interval for reconstruction based on EPS > 0.85, and there were 4 trees with 5 cores in 1760.

Point 3: Line 100 and/or Table 1 –RC chronology – RC abbreviation is not defined, it be should be explained

Response 3: Yes, I have added the definition of RC.

Point 4: Lines 107-114 - You give a brief description of the climatic conditions here, but there is no information about climatic data/ variables used in the analyses.

So, I have a questions: what climatic data were used in the study? What was the time range of these data? – period 1958-2013? It is not clear.

You show it in Figure 4 (section Results), but in my opinion these information should be in this section.

Response 4: Yes, you are right. I have added the information about climatic data used in the analyses. We employed average monthly temperature and total monthly precipitation (from previous July to current September) of the meteorological stations over the common period 1958-2013 to calibrate the ring-width climate relationships.

Point 5: Line 118- “The correlation analysis was applied to indicate the climate-tree growth linkages” – in my opinion this information is insufficient.

For example: no information about time interval for calculated the correlations coefficients, no information how was evaluated the statistical significance of the correlations, bootstrap methods? This information is necessary in this section.

The next sentence is very unclear (lines 118- 122).

Response 5: Yes, you are right. We use the simple Pearson correlation analysis in the SPSS program to indicate the climate-tree growth linkages. All statistical procedures were evaluated at a p < 0.05 level of significance.

Point 6: Lines 130-132 “We finally compared our precipitation reconstruction with previous streamflow reconstructions of the Yellow River to evaluate the impact of regional precipitation on streamflow” – no information about source this data

Moreover, information about the software used for calculations would also be useful.

Response 6: Yes, we have added references on the reconstruction of the Yellow River runoff.

Point 7: Lines 138-139 “Total August-June precipitation showed the highest significant positive correlation with RES chronology” - I have a question- Have you analysed the correlations for other periods? , for example, correlations between annual rainfall, or maybe other periods, for example, August-October of the previous year. If you have not analysed other periods, please write why you chose the August-June period.

Response 7: Yes, we have analyzed the correlations in other periods. In these periods, the total precipitation from August of the previous year to June of the current year has the highest correlation with the RES chronology.

Point 8: Figure 4 – please explain abbreviation P i C on the OX axis.

Response 8: I have added relevant instructions.

Point 9: Lines 162-166 “Low winter temperatures can lead to thicker snow cover, further delay initiation of growth in the following spring which reduced earlywood width [30]”. –Could you explain this sentence in relation to your results. After all, in your research correlations between winter temperature and temperature are negative.

Response 9: This is a mistake, we have corrected it. The low negative correlations between tree growth and mean temperature from prior July to current July reveal that temperature has no little influence on the tree growth at these sites.

Point 10: Linkage with steamflow of the Yellow River- typo error – should be streamflow

Response 10: Yes, I have corrected.

Reviewer 2 Report

The MS presents a classic dendroclimatological study, a reconstruction of high frequency precipitation from the North-eastern Tibetan plateau.

This is a region where the paucity of long instrumental temperature records complicates the assessment of climate change. The value of such datasets are of importance for characterisation of the temperature history in the study area. New reconstructions with a known climate signal are also important contributions to future studies that uses tree-ring networks.

I think the authors should be encouraged to expand and develop the methods and ideas to support the reliability of a new reconstruction. Below are some suggestions to increase the impact of the manuscript:

Major comments:

P3 L99-100: The authors refer to the EPS, SNR and MS. These are all useful descriptive statistics for comparing sites and species. They are measures of chronology reliability and coherence. However, EPS and SNR are both functions of the mean inter-series correlation and sample size, and thus cannot reveal the strength or nature of any environmental signal, nor be used to judge the dendroclimatic utility of the chronology. Please rephrase.

Also, the mean inter-series correlation is mentioned in the methods section, but the full results of that analysis is not presented in the MS. Please include Rbar changes over time e.g. in figure 2.

Since 3 sites are combined into one, it would also be beneficial to include the correlation between sites and the chronology characteristics for each site.

P3 L101 (figure 2): There seems to be some variance changes in the chronology that should be discussed along with their relevance to the reconstruction. Could they be related to the detrending procedure? The forest composition also deserves mention. Were the samples taken from open canopy forests or were the stands closed or clustered? This is important since it affects the type of climatic signal and disturbance regimes in the forest, and thus affects the quality of the signal. Additionally, it affects what detrending method may be chosen for the analysis. If the canopy is closed, then individual detrending seems proper, but fitting negative exponential functions to trees growing in clusters may not be proper (See Cook 1985; Cook and Peters 1981) and may introduce biases when the functions do not fit properly. 

Since the reconstruction targets only high-frequency climate signals, a more flexible detrending curve can well be applicable if the negative exponential has a poor fit in the very early/late parts of the series.

P4 L105: How was the Rbar and EPS calculated? Per core or per tree? With 127 cores from 54 trees, there are uneven numbers of cores per tree. Please note that calculating the EPS by core will inflate these statistics and might give an unrealistic assessment of the coherency between trees.

P4 L105: Please also include the mean segment length in the statistical properties table. With individual detrending, this is an important descriptive statistic to gauge the maximum resolvable low-frequency variability in the dataset (c.f. Cook et al. 1995).

P5 L146-150: The reconstruction targets August-June precipitation. However, from the correlation analysis, only the correlation to previous year’s September and October are statistically significant. The 55-year instrumental dataset should be just about long enough for a more rigorous verification procedure. I strongly encourage the authors to include the RE and CE form split-period verification. 

P6 L174-179: Is there autocorrelation in the regression residuals?

P8 L245: «…to emphasise long-term fluctuations.” Please rephrase, e.g. to “…empahsise decadal fluctuations” or something similar to avoid misunderstandings. The detrending method chosen prevents the detection of climate changes on timescales longer then the mean segment length.

P7/8 L228-241: the comparison to streamflow reconstruction is nice and it might be interesting to explore this in more detail. For example, applying moving correlations could help address periods with changes in the coherence between the records. When do they synchronise or deviate and under what kind of conditions? Does these periods relate to atmospheric circulation or monsoon strength?

Minor comments:

P1 L41-43: Please use the correct/traditional names and nomenclature of botanical taxa consistently. It is also customary to include the author citation in botanical nomenclature, i.e, Qinghai spruce (Picea crassifolia Kom.) and Qilian juniper (Juniperus przewalskii Kom.)

P3 L84: The map is very nice, but the labels of the sampling sites in the right hand panel are difficult to read.

P3 L90: TSAP-LINTAB, COFECHA and ARSTAN, please use the appropriate references for the software. 

P3 L87: Language. Please revise.

P3 L 94-96: Language. Please revise.

P4 figure 3: The degrees Celsius sign is incorrectly formatted in the right axis legend.

P6 L180: Please include a description of the meaning of the asterisks in the table.

Author Response

Point 1: P3 L99-100: The authors refer to the EPS, SNR and MS. These are all useful descriptive statistics for comparing sites and species. They are measures of chronology reliability and coherence. However, EPS and SNR are both functions of the mean inter-series correlation and sample size, and thus cannot reveal the strength or nature of any environmental signal, nor be used to judge the dendroclimatic utility of the chronology. Please rephrase.

Response 1: Yes, you are right, we have rephrased the sentence to “Statistical characteristics of the RC chronology show that a strong environmental influence on the growth of Picea crassifolia Kom.

Point 2: Also, the mean inter-series correlation is mentioned in the methods section, but the full results of that analysis is not presented in the MS. Please include Rbar changes over time e.g. in figure 2.

Response 2: Yes, I have added the statistical results of the chronology into the method section, and draw the statistical values of Rbar and EPS in Figure 2

Point 3: Since 3 sites are combined into one, it would also be beneficial to include the correlation between sites and the chronology characteristics for each site.

Response 3: Yes, I have added the correlation coefficient of raw ring-width data of three samples

Point 4: P3 L101 (figure 2): There seems to be some variance changes in the chronology that should be discussed along with their relevance to the reconstruction. Could they be related to the detrending procedure? The forest composition also deserves mention. Were the samples taken from open canopy forests or were the stands closed or clustered? This is important since it affects the type of climatic signal and disturbance regimes in the forest, and thus affects the quality of the signal. Additionally, it affects what detrending method may be chosen for the analysis. If the canopy is closed, then individual detrending seems proper, but fitting negative exponential functions to trees growing in clusters may not be proper (See Cook 1985; Cook and Peters 1981) and may introduce biases when the functions do not fit properly. Since the reconstruction targets only high-frequency climate signals, a more flexible detrending curve can well be applicable if the negative exponential has a poor fit in the very early/late parts of the series.

Response 4: You are familiar with tree-ring research in this area. As you know, the standard (STD) and ARSTAN (ARS) chronologies are usually used to reduce the possible effects of competition in closed canopy forests. In our study area, the tree stands are generally open, so that the possible effects of competition among trees are probably very low.

Point 5: P4 L105: How was the Rbar and EPS calculated? Per core or per tree? With 127 cores from 54 trees, there are uneven numbers of cores per tree. Please note that calculating the EPS by core will inflate these statistics and might give an unrealistic assessment of the coherency between trees.

Response 5: We calculated Rbar and EPS based on per core, and do not double-calculate.

Point 6: P4 L105: Please also include the mean segment length in the statistical properties table. With individual detrending, this is an important descriptive statistic to gauge the maximum resolvable low-frequency variability in the dataset (c.f. Cook et al. 1995).

Response 6: Yes, we have added the mean segment length to Table 1.

Point 7: P5 L146-150: The reconstruction targets August-June precipitation. However, from the correlation analysis, only the correlation to previous year’s September and October are statistically significant. The 55-year instrumental dataset should be just about long enough for a more rigorous verification procedure. I strongly encourage the authors to include the RE and CE form split-period verification.

Response 7: You are right, Split-sample calibration-verification tests were also employed to evaluate the statistical fidelity of our reconstruction model. The resulting statistics are shown in the flowing table. The reduction of error (RE) and the coefficient of efficiency (CE), are both positive, which also indicates significant skill in the tree ring estimates.

Calibration

(1959 – 1985)

Verification

(1986 – 2013)

Calibration

(1986 – 2013)

Verification

(1959 – 1985)

Reduction of Error

/

0.45

/

0.57

Coefficient of Efficiency

/

0.45

/

0.57

Point 8: P6 L174-179: Is there autocorrelation in the regression residuals?

Response 8: Yes, the correlation coefficient between the observed value and the predicted value is 0.687 (P < 0.01) during the calibration period.

Point 9: P8 L245: «…to emphasise long-term fluctuations.” Please rephrase, e.g. to “…empahsise decadal fluctuations” or something similar to avoid misunderstandings. The detrending method chosen prevents the detection of climate changes on timescales longer then the mean segment length.

Response 9: Yes, you are right, I have corrected.

Point 10: P7/8 L228-241: the comparison to streamflow reconstruction is nice and it might be interesting to explore this in more detail. For example, applying moving correlations could help address periods with changes in the coherence between the records. When do they synchronise or deviate and under what kind of conditions? Does these periods relate to atmospheric circulation or monsoon strength.

Response 10: We appreciate your suggestions and comments. The comparison between precipitation and runoff in the upper reaches of the Yellow River was also reported in other results [23]. The scientific understanding of the impact of precipitation on runoff is based on high-resolution precipitation and runoff data. In this article, we use total previous August-June precipitation reconstruction to compare the previous October- September flow, and the correlation coefficient is 0.34 (P < 0.01 1760-2011). If we have relevant monthly runoff data of the Yellow River, we will further analyze the impact of precipitation on runoff in more details.

Point 11: P1 L41-43: Please use the correct/traditional names and nomenclature of botanical taxa consistently. It is also customary to include the author citation in botanical nomenclature, i.e, Qinghai spruce (Picea crassifolia Kom.) and Qilian juniper (Juniperus przewalskii Kom.)

Response 11: Yes, you are right, and I have corrected.

Point 12: P3 L84: The map is very nice, but the labels of the sampling sites in the right hand panel are difficult to read.

Response 12: Yes, I have painted the labels of the sampling sites in the right hand panel in yellow.

Point 13: P3 L90: TSAP-LINTAB, COFECHA and ARSTAN, please use the appropriate references for the software.

Response 13: Yes, I have corrected.

Point 14: P3 L87: Language. Please revise.

Response 14: Yes, I have corrected.

Point 15: P3 L 94-96: Language. Please revise.

Response 15: Yes, I have corrected.

Point 16: P4 figure 3: The degrees Celsius sign is incorrectly formatted in the right axis legend.

Response 16: Yes, this is mistake, and have corrected.

Point 17: P6 L180: Please include a description of the meaning of the asterisks in the table.

Response 17: Yes, I have corrected.

Reviewer 3 Report

Interesting article, well written,

main remark: why there is no separate discussion - it is currently part of the results, e.g. 149-158, 161-166 and others
It is up to the editor to decide whether to leave the current form or transfer the discussion after results

line 20 and 24 - once the river is written with a capital letter and a lower case letter, make it uniform
line 40 is not dendrologists but dendrochronologists
line 73 which means: low precipitation cold wet mountains (low precipitation vs. wet ??)
figure 1 - mark research surfaces with a different color - now poorly visible, maybe black ??
Table 1 explain in the description what means MS, S / N, EPS, column with species not needed, which means RC in row 2
figure 3 what kind of unit is this at the temperature axis - poorly visible, improv
line 138 and further (e.g. 170,183, ...) - it is worth writing everywhere pVIII or previous August-June)
figure 4 which means P7 or C3 - explain, or write differently, e.g. pVII or pJUL
line 169 - why CE in 2013 ??
line 173 - here it is 51.4%, and in Figure 5a 52%
line 174 why actual August ??
table 2 - explain the abbreviations and what * and ** mean, RC = RCres ??
fig. 5a - r2 = 0.52 for sure ?? and not 0.51 ??
improve signature previous August - twice times
Fig. 6 - poor quality of the figure
Fig. 7 divided into A- flow .... and B - precipitation
what happens in the period 1925-1937 there is high rainfall and very low flows, how to explain it ??

Author Response

Point 1: line 20 and 24 - once the river is written with a capital letter and a lower case letter, make it uniform

Response 1: Yes, I have changed to capital letters uniformly.

Point 2: line 40 is not dendrologists but dendrochronologists

Response 2: Yes, I have corrected.

Point 3: line 73 which means: low precipitation cold wet mountains (low precipitation vs. wet ??)

Response 3: Yes, this is a mistake, I have deleted “wet mountains”.

Point 4: figure 1 - mark research surfaces with a different color - now poorly visible, maybe black ??

Response 4: Yes, I have painted the labels of the sampling sites in the right hand panel in yellow.

Point 5: Table 1 explain in the description what means MS, S / N, EPS, column with species not needed, which means RC in row 2

Response 5: Yes, I have added the explanation of the abbreviations (MS, S / N, EPS), and deleted the column with species. RC is a regional chronology which combined all raw ring-width data from three sampling sites.

Point 6: figure 3 what kind of unit is this at the temperature axis - poorly visible, improve

Response 6: Yes, I have corrected.

Point 7: line 138 and further (e.g. 170,183, ...) - it is worth writing everywhere pVIII or previous August-June)

Response 7: Yes, you are right, I have changed all of them to previous August-June.

Point 8: figure 4 which means P7 or C3 - explain, or write differently, e.g. pVII or pJUL

Response 8: Yes, I have added explanations of P and C in the title of figure 4.

Point 9: line 169 - why CE in 2013 ??

Response 9: Because the tree-ring sampling points MAX and MXS were collected in the summer of 2014.

Point 10: line 173 - here it is 51.4%, and in Figure 5a 52%

Response 10: Yes, we change 52% in Figure 5a to 51.4%.

Point 11: line 174 why actual August ??

Response 11: We have corrected it to “actual previous August-June”.

Point 12: table 2 - explain the abbreviations and what * and ** mean, RC = RCres ??

Response 12: Yes, we have added the explanation.

Point 13: fig. 5a - r2 = 0.52 for sure ?? and not 0.51 ?? improve signature previous August - twice times

Response 13: Yes, we change 52% in Figure 5a to 51.4%.

Point 14: Fig. 6 - poor quality of the figure

Response 14: Yes, I have corrected.

Point 15: Fig. 7 divided into A- flow .... and B – precipitation

Response 15: Yes, I have corrected.

Point 16: what happens in the period 1925-1937 there is high rainfall and very low flows, how to explain it ??

Response 16: Documents show that in the summer of 1928 CE, flow in the main Yellow River tributaries, such as the Jinghe and Weihe Rivers, was completely cut off, and horses and chariots could traverse dry river beds [25].

Round 2

Reviewer 2 Report

I only have a few comments before recommending this MS for publication: 

P1L41&43: in latin species names, the author citation should not be italicised, only the species epithet.

P3L 110-111: The sentence is unclear.

P3L 104-106: Since the EPS and Rbar were calculated in in a way that might inflate the numbers i.e. on cores rather than trees, it should be stated in the manuscript how it was done.

P3L99: I refer to the previous comment:

Point 4: P3 L101 (figure 2): There seems to be some variance changes in the chronology that should be discussed along with their relevance to the reconstruction. Could they be related to the detrending procedure? The forest composition also deserves mention. Were the samples taken from open canopy forests or were the stands closed or clustered? This is important since it affects the type of climatic signal and disturbance regimes in the forest, and thus affects the quality of the signal. Additionally, it affects what detrending method may be chosen for the analysis. If the canopy is closed, then individual detrending seems proper, but fitting negative exponential functions to trees growing in clusters may not be proper (See Cook 1985; Cook and Peters 1981) and may introduce biases when the functions do not fit properly. Since the reconstruction targets only high-frequency climate signals, a more flexible detrending curve can well be applicable if the negative exponential has a poor fit in the very early/late parts of the series.

Response 4: You are familiar with tree-ring research in this area. As you know, the standard (STD) and ARSTAN (ARS) chronologies are usually used to reduce the possible effects of competition in closed canopy forests. In our study area, the tree stands are generally open, so that the possible effects of competition among trees are probably very low.

I am not talking about autoregressive modelling here but of whether your choice of the negative exponential curve is a good fit to the data. (By the way, the STD chronology is the detrended chronology, the RES are the residuals after univariate autoregressive modelling on the STD, and the ARS chronology, which is Cook’s attempt to maximise the climate signal, is the only chronology where the pooled autoregression residuals have been added back in. It is therefore more correct to say that ARS is the attempt to maximise the climate variation. If you have the strongest climate correlation in the RES chronology, it probably means that the climate variation in your dataset is only in the high frequencies. Did you attempt to use STD or ARS? They would preserve more low-frequency variance than RES and could provide a better reconstruction also in open-canopy forests.)

But using the residuals (RES) is not a problem in itself and the reason for discussing this was the variance changes. Since the canopy is open in this case and you do not mention improper curve fits, could rather the variance changes in your chronology be related to the introduction of many young and short-lived trees around 1950?

Please mention in your paper that the canopy is open and your interpretation of whether this variance change is likely related to climate or could be an artefact of chronology composition. It is important for those trying to interpret the reconstruction.

P6 L188-197: The results of split period verification is not included in the MS. I strongly encourage the authors to include these statistics as they are important in the worldwide evaluation of tree-ring divergence.

Author Response

Point 1: P1L41&43: in latin species names, the author citation should not be italicised, only the species epithet.

Response 1: Yes, I have corrected.

Point 2: P3L 110-111: The sentence is unclear.

Response 2: Yes, I have corrected.

Point 3: P3 L104-106: Since the EPS and Rbar were calculated in in a way that might inflate the numbers i.e. on cores rather than trees, it should be stated in the manuscript how it was done.

Response 3: Yes, I have added the relevant sentence.

Point 4: P3L99: I refer to the previous comment: I am not talking about autoregressive modelling here but of whether your choice of the negative exponential curve is a good fit to the data. (By the way, the STD chronology is the detrended chronology, the RES are the residuals after univariate autoregressive modelling on the STD, and the ARS chronology, which is Cook’s attempt to maximise the climate signal, is the only chronology where the pooled autoregression residuals have been added back in. It is therefore more correct to say that ARS is the attempt to maximise the climate variation. If you have the strongest climate correlation in the RES chronology, it probably means that the climate variation in your dataset is only in the high frequencies. Did you attempt to use STD or ARS? They would preserve more low-frequency variance than RES and could provide a better reconstruction also in open-canopy forests.)

But using the residuals (RES) is not a problem in itself and the reason for discussing this was the variance changes. Since the canopy is open in this case and you do not mention improper curve fits, could rather the variance changes in your chronology be related to the introduction of many young and short-lived trees around 1950?

Response 4: We appreciate your suggestions and comments. The negative exponential curve fits were used to remove nonclimatic trends also reported around the study area [16, 23-24]. And we tried to use STD chronology and RES chronology to do correlation analysis with previous July- September meteorological data. After combining the months, total previous August-June precipitation showed the highest significant positive correlation with STD chronology (r = 0.67, P < 0.01) and RES chronology (r = 0.72, P < 0.01), respectively. In addition, after detrending with the negative exponential curve, all detrended series were averaged to chronologies by computing the biweight robust mean in the ARSTAN program.

Point 5: Please mention in your paper that the canopy is open and your interpretation of whether this variance change is likely related to climate or could be an artefact of chronology composition. It is important for those trying to interpret the reconstruction.

Response 5: Yes, I have added the relevant sentence.

Point 6: P6 L188-197: The results of split period verification is not included in the MS. I strongly encourage the authors to include these statistics as they are important in the worldwide evaluation of tree-ring divergence.

Response 6: Yes, I have added the results of split period verification.

This manuscript is a resubmission of an earlier submission. The following is a list of the peer review reports and author responses from that submission.